# Diversity and Antimicrobial Resistance in the *Streptococcus bovis*/*Streptococcus equinus* Complex (SBSEC) Isolated from Korean Domestic Ruminants

**DOI:** 10.3390/microorganisms9010098

**Published:** 2021-01-04

**Authors:** Seon Young Park, Mingyung Lee, Se Ra Lim, Hyemin Kwon, Ye Seul Lee, Ji Hyung Kim, Seongwon Seo

**Affiliations:** 1Infectious Disease Research Center, Korea Research Institute of Bioscience and Biotechnology, Daejeon 34141, Korea; lovesun139@kribb.re.kr (S.Y.P.); seraxx@naver.com (S.R.L.); hena0922@kribb.re.kr (H.K.); znsdjssl@naver.com (Y.S.L.); 2Division of Animal and Dairy Sciences, College of Agriculture and Life Science, Chungnam National University, Daejeon 34134, Korea; mingyung1203@cnu.kr

**Keywords:** *Streptococcus bovis*, bovine acidosis, genome, *tet(M)*-possessing tn916-like transposon

## Abstract

*S. bovis*/*S. equinus* complex (SBSEC) includes lactic acid-producing bacteria considered as the causative agent associated with acute rumen lactic acidosis in intensive ruminants. Considering the limited information on the detailed characteristics and diversity of SBSEC in Korea and the emergence of antimicrobial resistance (AMR), we investigated the diversity of SBSEC from domestic ruminants and verified the presence of antimicrobial resistance genes (ARGs) against several antimicrobials with their phenotypic resistance. Among 51 SBSEC isolates collected, two SBSEC members (*S. equinus* and *S. lutetiensis*) were identified; *sodA*-based phylogenetic analyses and comparisons of overall genome relatedness revealed potential plasticity and diversity. The AMR rates of these SBSEC against erythromycin, clindamycin, and tetracycline were relatively lower than those of other SBSEC isolates of a clinical origin. An investigation of the ARGs against those antimicrobials indicated that tetracycline resistance of SBSECs generally correlated with the presence of *tet(M)*-possessing *Tn916-like* transposon. However, no correlation between the presence of ARGs and phenotypic resistance to erythromycin and clindamycin was observed. Although a limited number of animals and their SBSEC isolates were examined, this study provides insights into the potential intraspecies biodiversity of ruminant-origin SBSEC and the current status on antimicrobial resistance of the bacteria in the Korean livestock industry.

## 1. Introduction

The intensive management systems in the current livestock industry encourage the use of high-concentrate diets rather than high-forage diets in ruminants to enhance productivity and cost efficiency [1,2,3]. The rumen-bacterial community largely depends on the type of feed ingested by the host ruminants and rapid changes in the rumen bacterial community can significantly affect animal health and productivity [4,5]. Feeding ruminants with high-concentrate diets increases the level of non-fibrous carbohydrates, which promotes the proliferation of lactic acid producing amylolytic bacteria (e.g., *Streptococcus bovis* and *Lactobacillus* spp.) [6,7], with the accumulation of lactate ultimately leading to a rapid decrease in ruminal pH, causing acute rumen acidosis [8,9]. Rumen acidosis is reported to be associated with several clinical signs including intake depression, reduced fiber digestion, milk fat depression, diarrhea, ruminitis, lameness, liver abscesses, inflammation, pneumonia, and even death and thus, it is considered one of the most important metabolic disorders in intensive ruminants [10].

*Streptococcus* (*S.*) *bovis*, synonymized with *S. equinus* and currently recognized as *S. bovis*/*S. equinus* complex (SBSEC), is commonly found in the digestive tract of humans and ruminants. In addition to ruminants and humans, SBSEC have also been reported in companion animals, livestock (poultry and pigs), and their food products, as well as in wild animals (birds, marsupials, and aquatic mammals) [11,12]. Previously, the SBSEC have been divided into three biotypes designated as biotype I, biotype II/1, and biotype II/2 however, the taxonomy of the SBSEC has undergone several taxonomical changes over the past 20 years in accordance with the description of new species (or subspecies) originally grouped as *S. bovis* [13,14]. The most recent taxonomical revision based on genetic biomarkers (e.g., 16S rRNA, *sodA*, and *groEL*) describes SBSEC as comprising seven species (or subspecies) including *S. equinus*, *S. lutetiensis* (previously *S. infantarius* subsp. *coli*), *S. infantarius* subsp. *infantarius*, *S. alactolyticus*, *S. gallolyticus* subsp. *pasteurianus*, *S. gallolyticus* subsp. m*acedonicus*, and *S. gallolyticus* subsp. *galloyticus* [15]. Although most SBSEC have been described as commensal bacteria, some members are associated with infective endocarditis and colorectal cancer in humans and animals however, the mechanisms by which these members shift from commensal organisms to pathogens remain unclear [16].

In ruminants, SBSECs produce lactic acid when they grow rapidly with sufficient amounts of non-fibrous carbohydrates and are considered one of the important causative agents associated with rumen acidosis [9,11,17]. Recent studies on the rumen microbiome in cattle have provided strong evidence that the SBSEC is an important contributor to ruminal acidosis [18,19] and given the potential causative role of bacteria, numerous strategies to specifically prevent their overgrowth in feedlot cattle have been reported [9]. Interestingly, the physiological and genetic diversities of bovine-origin SBSEC have been addressed [20,21] and genetic heterogeneity in SBSEC, especially in *S. equinus* isolates, have been described recently [22]. However, the potential plasticity and diversity of SBSEC in domestic ruminants has not been well investigated so far.

Antimicrobials agents (or antibiotics) have been used in the global livestock industry for the prevention and control of diseases, as well as growth promoters however, the emergence of antimicrobial resistance (AMR) and widespread antimicrobial resistance genes (ARGs) have led to serious industrial and public health concerns due to potential health risks to humans and animals [23,24]. Several classes of antimicrobial agents (e.g., tetracyclines, quinolones, aminoglycosides, and macrolides) are widely used in domestic animals and the dissemination of ARGs associated with these antibiotics (e.g., the *erm*, *tetR*, and *qnr* clusters) has been frequently reported from different geographical regions [25,26]. Previous studies have also described the emergence of AMR in some SBSEC strains in livestock products [27,28,29]. However, information regarding the recent prevalence of AMRs in ruminal SBSEC isolates in the global livestock industry is limited compared with that in human medicine [15].

Although the rumen bacterial diversity of Korean domestic ruminants has been investigated [30,31], the diversity and potential AMRs of SBSEC in these animal species have not been elucidated. Therefore, in this study, we investigated the diversity of SBSEC isolates from three Korean domestic ruminant species: Hanwoo steers (*Bos taurus coreanae*), Holstein dairy cattle (*Bos taurus*), and Korean native goats (*Capra hircus coreanae*). We also verified the presence of ARGs associated with several classes of antimicrobials that are widely used in domestic ruminants with their phenotypic resistance. To the best of our knowledge, this is the first report to assess the diversity of SBSEC from Korean domestic ruminants.

## 2. Materials and Methods

### 2.1. Animal Care

This study was conducted at the Center for Animal Science Research, Chungnam National University, Korea. The animal use and protocols for this experiment were reviewed and approved by the Chungnam National University Animal Research Ethics Committee (IACUC approval Nos. CNU-00455 (April, 2014) and CNU-01021 (April, 2018)).

### 2.2. Collection of Rumen Fluids

The rumen fluid samples of bovine (Holstein dairy cattle and Hanwoo) and caprine (Korean native goat) individuals reared in the Center for Animal Science Research, Chungnam National University (Korea), were collected via an oral stomach tube or a rumen cannula. First, the rumen fluids of 4 non-pregnant, non-lactating Hanwoo cows (546 ± 33.6 kg), 4 non-pregnant, non-lactating Holstein dairy cows (516 ± 42.7 kg), and 4 Korean native goats (19 ± 1.4 kg), individually housed, were collected via an oral stomach tube as previously described [31]. In addition, the rumen fluid samples of 2 individually housed non-pregnant and non-lactating Holstein dairy cows fitted with a permanent fistula (698 ± 148.5 kg) were collected. The animals were fed 700 g/kg of corn silage and 300 g/kg of commercial concentrate mix twice a day, and the chemical compositions of the corn silage and concentrate mix are presented in Appendix A. Drinking water was provided ad libitum to the animals throughout the experimental period. The collected rumen fluids were immediately transferred to the laboratory for the isolation of SBSEC.

### 2.3. Bacterial Isolation and Identification

Sterile swabs were used to collect specimens from the rumen fluid samples. Bacteria were isolated using the standard dilution plating technique on the agar media with two different compositions of carbohydrates containing clarified rumen fluids [30,32] and Brain Heart Infusion agar (BHIA; Difco, Detroit, MI), followed by incubation at 37 °C for 24 h. A total of 5 white or orange-colored opaque colonies on each plate were randomly selected and subcultured three times. Then, the isolated bacteria were preferentially identified using the 16S rRNA sequencing analysis. Bacterial genomic DNA was isolated using the DNeasy Blood & Tissue Kit (Qiagen Korea Ltd., Seoul, Korea), according to the manufacturer’s protocol. The 16S rRNA gene was amplified using universal primers 27F and 1492R, and the amplicons were sequenced using universal primers 785F and 907R (Appendix A). Overall, the PCR amplifications in this study were performed using the Maxime PCR PreMix kit (Intron Biotechnology, Seongnam, Korea), and all PCR products were purified using the QIAquick Gel Extraction Kit (Qiagen Korea Ltd., Seoul, Korea) before sequencing by Macrogen Inc. (Seoul, Korea). To exclude the repeated isolation of the same bacterial strain, the isolates sharing 100% 16S rRNA identity from the same rumen fluid sample were considered as a single strain in this study. Thereafter, the biochemical characteristics of isolates identified as members of the genus *Streptococcus* were analyzed using the API 20 Strep system (bioMérieux Inc., Marcy l’ Étoile, France) following the manufacturer’s protocol. All confirmed *Streptococcus* isolates were stored in Brain Heart Infusion broth (Difco, Detroit, MI) supplemented with 10% glycerol at −80 °C until use.

### 2.4. Species Discrimination and Phylogenetic Analysis

The obtained *Streptococcus* isolates were cultured overnight on BHIA at 37 °C. For species discrimination, the *sodA* gene, encoding the manganese-dependent superoxide dismutase, was amplified and sequenced using the primers d1/d2 (Appendix A) [33,34]. The amplified and sequenced *sodA* gene of the isolates were respectively compared with other SBSEC strains including the type strains in the GenBank database by BLAST searches (www.ncbi.nlm.nih.gov/BLAST). Furthermore, the *sodA* sequences of the isolates were aligned with representative sequences from each SBSEC type strain using ClustalX (version 2.1) [35] and BioEdit Sequence Alignment Editor (version 7.1.0.3) [36]. The datasets were then analyzed phylogenetically using MEGAX (ver. 10.0) [37]. Phylogenic trees were constructed using the Neighbor-Joining (NJ) and Maximum-Likelihood (ML) methods, and the reliability of the trees was assessed using 1000 bootstrap replicates.

### 2.5. Antimicrobial Susceptibility Test

Antimicrobial susceptibility of the isolated SBSEC bacteria was evaluated using the disk diffusion method according to the guidelines of the Clinical and Laboratory Standards Institute (CLSI) [38,39,40]. In total, 12 antimicrobial agents (Oxoid Ltd., Basingstoke, UK) from 11 classes were used as follows: Aminoglycosides [gentamicin (10 μg)], carbapenems [imipenem (10 μg)], cephalosporins [cephalothin (30 μg)], fluoroquinolones [levofloxacin (5 μg)], glycopeptides [vancomycin (30 μg)], lincosamides [lincomycin (15 μg)], macrolides [erythromycin (15 μg)], oxazolidinones [linezolid (30 μg)], penicillins and β-lactam/β-lactamase inhibitor combinations [oxacillin (1 μg), penicillin (10 μg)], phenicols [chloramphenicol (30 μg)], and tetracyclines [tetracycline (30 μg)]. The minimum inhibitory concentrations (MICs) of four selected antimicrobial agents [gentamicin (256–0.016 μg), erythromycin (256–0.016 μg), clindamycin (256–0.016 μg), and tetracycline (256–0.016 μg)] were determined using MIC Evaluator Strips (Liofilchem^®^, Teramo, Italy). Standard disk diffusion and MIC tests were conducted on the Muller-Hilton Blood Agar (Synergy Innovation, Seongnam, Korea) at 37 °C for 24 h and the isolates were categorized as susceptible, intermediate, and resistant based on the interpretive criteria of the CLSI guidelines [40,41,42]. For quality control, *Streptococcus pneumoniae* ATCC 49619 were used.

### 2.6. Genome Sequencing of the SBSEC Isolates

Based on the phylogenetic analysis, a total of 5 representative SBSEC isolates were selected and submitted for further genomic investigations. The genomes of the selected SBSEC strains (CNU_77-23, CNU_77-61, CNU_G2, CNU_G3, and CNU_G6) were sequenced using a hybrid approach on a PacBio RS II system (Pacific Biosciences) by constructing a 20-kb SMRTbellTM template library and on the HiSeq X-10 platform (Illumina) by preparing a DNA library using the TruSeq nano DNA library prep kit (Illumina). Genome assembly of the filtered PacBio reads was performed using the HGAP (v3.0) pipeline, and the Illumina paired-end 150-bp reads were mapped using BWA-MEM (v0.7.15) and errors were corrected using Pilon (v1.21) with default parameters. Annotation was performed with the NCBI Prokaryotic Genome Annotation Pipeline (http://www.ncbi.nlm.nih.gov/books/NBK174280/). The *L-lactate dehydrogenase* gene, which is responsible for lactic acid production, was manually searched in the sequenced SBSEC genomes by BLAST (www.ncbi.nlm.nih.gov/BLAST). Moreover, the presence of genetic determinants related to antibiotic resistance was screened using the Comprehensive Antibiotic Resistance Database (https://card.mcmaster.ca/) and/or the ARG-ANNOT database (http://en.mediterranee-infection.com/article.php?laref=283&titre=arg-annot). Finally, to assess the genomic relatedness of the five selected SBSEC isolates with other *Streptococcus* species in the SBSEC, the average nucleotide identity (ANI) was analyzed using OrthoANI (https://www.ezbiocloud.net/tools/othoani) against several other type strains of species (or subspecies) available in the GenBank database.

### 2.7. Determination of Antimicrobial-resistant Genes and L-lactate Dehydrogenase Genes in the SBSEC Isolates

The presence of genetic determinants conferring resistance to macrolides, lincosamides, and tetracyclines in the SBSEC isolates were investigated by PCR and sequencing analyses. Detections of resistance genes for macrolides [*erm(A)*, *erm(B)*, *erm(C)*, and *mef(A)*], lincosamides [*lnu(C)*], and tetracyclines [the putative conjugative Tn916-like transposon and *tet(M)*] were determined by conventional PCR and those for *tet(O)*, *tet(Q)*, and *tet(S)* were determined by multiplex PCR using previously described primers and protocols [33,41,42,43,44,45,46]. Moreover, the presence of genetic determinants responsible for lactic acid production (*L-lactate dehydrogenase*, *ldh*) was analyzed by PCR and sequencing analyses. The primer pair for the PCR amplification was generated from the annotated *ldh* genes encoded in the genome of the *S. equinus* strain CNU_G6 (GenBank accession No. CP046629; *ldh* region 1571618–1572607) using Primer 3 (http://bioinfo.ut.ee/primer3-0.4.0/primer3/). The primer sequences, amplicon sizes, and annealing temperatures used are summarized in Appendix A. Strains yielding amplicons of the expected size were sequenced and the obtained nucleotide and deduced amino acid sequences were compared to those of genes from other *Streptococcus* spp. including SBSEC, available in the GenBank database.

### 2.8. Accession Numbers of Nucleotide Sequences and Strain Deposition

All the 16S rRNA, *sodA*, *tet(M)*, *lnu(C)*, and putative conjugative Tn916-like transposon genes of the SBSEC isolates in this study have been deposited in the GenBank database and the accession numbers are provided in Table 1. The complete genome sequences of the representative SBSEC isolates have been deposited in GenBank database under accession numbers CP046628 (CNU_77-23), CP046875 (CNU_77-61), CP046919 (CNU_G2), CP046624 (CNU_G3), and CP046629 (CNU_G6), respectively. A living axenic culture of the 51 SBSEC isolates has been deposited in the Korean Culture Center of Microorganisms (KCCM) and the deposition numbers are provided in Table 1.

## 3. Results

### 3.1. SBSEC Isolation and Identification

Among the bacteria collected during the surveillance studies between 2014 and 2019, a total of 51 isolates from the rumen fluid of Holstein dairy cattle (*n* = 15), Hanwoo steers (*n* = 17), and Korean goats (*n* = 19), were identified as *Streptococcus* spp. based on 16S rRNA sequencing analysis. The biochemical characteristics of the bacterial isolates were analyzed using the API 20 Strep system (bioMérieux Inc., Marcy l’ Étoile, France) and the results are summarized in Appendix A. Briefly, all the isolates showed positive results for leucine aminopeptidase, raffinose, amidon, and glycogen, and most isolates (except strain CNU_77-2) were positive for lactose fermentation, which produces lactate.

During the bacterial identification in this study, the taxonomical positions of the 51 *Streptococcus* spp. isolates were not clearly discriminated by 16S rRNA sequence analyses therefore, the *sodA* gene, reported as one of the most reliable biomarkers for identifying SBSEC species [14,47], was used for species classifications in this study. Accordingly, all 51 isolates were confirmed to belong to the SBSEC and a total of two *Streptococcus* species including *S. equinus* (*n* = 46) and *S. lutetiensis* (*n* = 5) were identified. From the Hanwoo and Holstein dairy cattle, a total of 32 *S. equinus* strains (24 and eight each) were identified whereas a total of 14 *S. equinus* and five *S. lutetiensis* strains were obtained from Korean native goats (Table 1). The determined 16S rRNA and *sodA* sequences of the 51 SBSEC strains were deposited in the GenBank database and are listed in Table 2. Phylogenetic analyses based on the obtained *sodA* genes using the ML method revealed that the 46 *S. equinus* isolates were divided into four major types (type I, II, III, and IV) and two individual strains (CNU_G2 and CNU_G5), which were not clustered with the other isolates (Figure 1). Among the four major types, the reference strains of *S. equinus* (ATCC 9812^T^ (Z95903) and ATCC 33317^T^ (AY344537), which were previously reported as *S. bovis*) were assigned to type IV by clustering only with some other bovine-originated SBSEC isolates. Moreover, five other isolates, which were obtained from caprine animals and classified as *S. lutetiensis* (CNU_33, CNU_77-61, CNU_77-62, CNU_77-64, and CNU_77-76), were well clustered with the type strain of the species, *S. lutetiensis* NEM 782^T^ (AJ297189). A similar result was also obtained from the phylogenetic tree generated using the NJ method.

### 3.2. Antimicrobial Susceptibility of the SBSEC Isolates

The resistance profiles of the 51 SBSEC isolates to several antibiotic classes was evaluated using the disc diffusion and MIC methods based on the CLSI guidelines. In this study, none of the SBSEC isolates were resistant to oxacillin, penicillin, cephalothin, and imipenem furthermore, they were generally susceptible (including intermediate) to aminoglycosides (gentamicin, resistance rate 21.6% (11/51)), fluoroquinolones (levofloxacin, resistance rate 23.5% (12/51)), macrolides (erythromycin, resistance rate 2.0% (1/51)), lincosamides (clindamycin, resistance rate 0% (0/51)), and tetracycline (tetracycline, resistance rate 15.7% (8/51)) (Table 3). However, large proportions of SBSEC isolates were categorized as ‘non-susceptible’ against glycopeptides (vancomycin, non-susceptible rate 84.3% (43/51)) and oxazolidinones (linezolid, non-susceptible rate 94.1% (48/51)) according to the CLSI guidelines [38,39]. Although a total of 12 SBSEC strains (23.5%) were not resistant to any of the antimicrobials used in this study, all isolates were not susceptible to at least one antimicrobial agent, and 25.5% of the isolates (13/51) were resistant to more than two antimicrobial agents tested in this study. Moreover, 5.9% of the isolates (3/51) were resistant to three different classes of antimicrobial agents and two isolates (CNU_77-23 and CNU_77-43) were resistant to four different classes of antimicrobial agents. The overall antimicrobial resistance profiles of all SBSEC isolates tested are summarized in Table 3.

### 3.3. Genomic Features of the SBSEC Isolates

In this study, we selected and sequenced the whole genome of four representative *S. equinus* isolates (CNU_G3 which grouped in type II; CNU_G6 which clustered with ATCC 33317^T^, that was previously reported as *S. bovis* in type IV; CNU_77-23 which clustered with the type strain of *S. equinus* (ATCC 9812^T^) in type IV; and CNU_G2 which was not clustered with any other *S. equinus* strains used in this study) for the following reasons: First, to elucidate their plasticity and diversity in the *S. equinus* group, and second, to obtain the reference genomes to determine the genetic determinants related to the production of lactic acid and antimicrobial resistances that could be used for further overall screening in the 51 SBSEC isolates. Additionally, one strain of *S. lutetiensis* isolated from caprine (CNU_77-61) was also chosen and submitted for further genome sequencing and analysis.

The genomes of the five selected isolates comprised 1.9–2.0 Mbp, consisting of one chromosome each (1.8–1.9 Mbp with 37.4–37.9 G + C content (%)). Although the presence of plasmids in the SBSEC is known to be relatively rare [11], the three strains (CNU_G2, CNU_G3, and CNU_G6) identified as *S. equinus* possessed a single plasmid, but no antibiotic-resistant or virulence-related genes were detected in it. Detailed features of the sequenced genome and plasmids are indicated in Table 4. Genomic similarities among the five selected isolates and other available type strains of the SBSEC were assessed using the ANI comparisons and the results indicated that the genome of strains CNU_77-23 and CNU_G6 were the most similar to that of *S. bovis* ATCC 33317^T^ with > 96.0% ANI values (Figure 2a). However, *S. equinus* strains CNU_G2 and CNU_G3 and the *S. lutetiensis* strain CNU_77-61 showed <95.0% ANI values against any of the type strains of the SBSEC including *S. equinus* ATCC 9812^T^ and *S. lutetiensis* NCTC 13774^T^. Moreover, the resultant phylogeny based on the OrthoANI values also indicated that these three SBSEC strains were clearly differentiated from the type strains of *S. equinus* and *S. lutetiensis*, and were rather closely related to the available genome of *S. infantarius* subsp. *infantarius* (GenBank accession No. NZ_ABJK02000000) (Figure 2b). Although the three SBSEC isolates were identified as *S. equinus* and *S. lutetiensis* based on the 16S rRNA and *sodA* gene analyses, these results strongly indicated that the three SBSEC strains (CNU_G2, CNU_G3, and CNU_77-61) could not be classified as *S. equinus* and *S. lutetiensis*, and might be categorized as potential new species (or subspecies) in the SBSEC.

### 3.4. Antimicrobial-Resistant Genes and L-lactate Dehydrogenase Genes in the SBSEC Isolates

The presence of genetic determinants conferring resistance to macrolides, lincosamides, and tetracyclines in the five genome-sequenced SBSEC isolates (CNU_77-23, CNU 77-61, CNU_G2, CNU_G3, and CNU_G6) was preliminarily screened by manual searches using available databases. As a result, the lincosamide-resistance gene, *lnu(C)*, was found to be encoded in the bacterial chromosome of strain CNU_77-23 and CNU_G3 however, none of the genomes showed the known AMR genes for macrolides [*erm(A)*, *erm(B)*, *erm(C)*, and *mef(A)*]. Interestingly, the putative conjugative Tn916-like transposon that harbored the tetracycline-resistance gene, *tet(M)*, was encoded in the bacterial chromosome of all the sequenced SBSEC isolates except CNU_G3 and the Tn916-like transposons (~18 kb) found in the SBSEC genomes were almost identical (>99%) to those found in the *Enterococcus faecalis* DS16 transposon Tn916 (U09422.1; 18,032 bp), and its implicated *tet(M)* genes in the isolates were also very similar (>99%) to those from various G(+) bacterial species available in the GenBank database. Therefore, we screened the presence of *lnu(C)*, *erm(A)*, *erm(B)*, *erm(C)*, *mef(A)*, *tet (O)*, *tet (Q)*, *tet (S)*, and Tn916-like transposon including the *tet(M)* gene in the remaining 46 isolates and the results indicated that 20.0% (10/51) and 33.3% (17/51) of the SBSEC isolates possessed *lnu(C)* and the Tn916-like transposon including the *tet(M)* gene, respectively. Among these, a total of three isolates (CNU_77-23, CNU_GF, and CNU_G1) possessed both the *lnu(C)* and *tet(M)* genes, however, *erm(A)*, *erm(B)*, *erm(C)*, *mef(A)*, *tet (O)*, *tet (Q)*, and *tet (S)* were not detected in any of the SBSEC isolates. Interestingly, all five *S. lutetiensis* isolates (CNU_33, CNU_77-61, CNU_77-62, CNU_77-64, and CNU_77-76) obtained from caprine possessed the Tn916-like transposon but *tet(M)* was not found.

We also assessed the presence of the *ldh* gene related to lactic acid production, in the sequenced SBSEC genomes, and found it to be encoded in the chromosome of all the sequenced SBSEC genomes. The annotated *ldh* genes (990 bp) in the SBSEC isolates were respectively compared with the fructose-1,6-diphosphate-dependent L-lactate dehydrogenase in *S. equinus* JB1 (U60997.1) [48], and the nucleotide sequence (99.1–99.8% identities) and deduced amino acid sequence (99.7–100% identities) of these *ldh* genes were almost homologous to those of strain JB1. Therefore, we screened the presence of the *ldh* gene in the remaining 46 isolates using ldh-F/R PCR primers designed in this study, and the results indicated that all the 51 SBSEC strains possessed *ldh*, suggesting that our isolates may have a strong potential to produce lactic acid as was shown in the results of biochemical tests using the API 20 Strep system.

## 4. Discussion

For several decades, the SBSEC has been recognized as commensal bacteria that inhabit the gastrointestinal tract of ruminants and humans, but numerous studies have provided evidence that the complex is an important contributor to ruminal acidosis in domestic ruminants and are also associated with serious infections and colorectal cancer in humans and animals, thus highlighting the common importance of SBSEC members in the livestock industry and human health [15,16]. Although the exact economic damage caused by SBSEC in domestic animals is not currently clear, it is generally understood that the bacteria can affect animal health, productivity leading to important losses in the livestock industry [9]. Along with ruminants, SBSEC have been reported from various other domestic animals, their food products, and from companion wild animals however, only limited numbers of prevalence studies have been conducted on birds, cattle, and lambs [16].

Although *S. bovis* was synonymized with *S. equinus* and is currently recognized as the SBSEC, the name is still commonly used in ruminant livestock research [49,50,51]. However, researchers in the area have already recognized the physiological and genetic diversities of bovine-origin *S. bovis* isolates [20,21] and genetic heterogeneity in SBSEC has been described recently, especially in *S. equinus* isolates [22]. Nevertheless, the potential plasticity and diversity of SBSECs in domestic ruminants have not been well investigated so far. In this milieu, we preferentially aimed to investigate the prevalence and diversity of SBSEC inhabiting Korean domestic ruminants.

In this study, a total of 51 SBSEC strains (Holstein dairy cattle (*n* = 15), Hanwoo (*n* = 17), and Korean goats (*n* = 19)) were isolated from collected rumen fluids. As expected, most of the isolates (except strain CNU_77-2) were positive for lactose fermentation which produces lactate, thus suggesting that these isolates have strong potential to cause ruminal acidosis in animals. However, no significant differences in biochemical characteristics between isolates from the three animal species was observed except for esculin, which was positive only in some isolates (*n* = 5) from Hanwoo steers (Appendix A). Esculin hydrolysis has been considered an important diagnostic factor to differentiate enterococci and group D streptococci (mainly SBSEC) from non-group D viridans group streptococci [52] and several SBSEC strains have been identified successfully using esculin hydrolyzing activity [53]. However, based on the results of our study, the SBSEC strains isolated from Korean domestic ruminants seem to have relatively low esculin hydrolyzing activity (9.8%), suggesting that the bile-esculin test might not be suitable for the presumptive identification of SBSEC, at least from Korean domestic ruminants.

For species discrimination of the 51 SBSEC strains collected in this study, both the 16S rRNA and *sodA* genes were respectively obtained and compared with the seven-type-SBSEC strains. In accordance with previous reports [14,47,54,55,56], partial *sodA* gene sequencing was more discriminant than 16S rRNA for SBSEC species identification in this study. Therefore, the species of the 51 SBSEC isolates were finally determined by *sodA* gene-based identification, and as a result, all 51 isolates were confirmed to belong to SBSEC, with a total of two *Streptococcus* species including *S. equinus* (*n* = 46) and *S. lutetiensis* (*n* = 5) identified. Interestingly, *S. lutetiensis* was only isolated from the rumen fluids of goats rather than those from dairy cattle and Hanwoo in this study. Although it is difficult to directly compare the presence of SBSEC members in a single animal species due to the difficulties encountered over the years in the correct identification of SBSEC strains to the species (or subspecies) level by phenotypic and genotypic methods, the potential presence of *S. lutetiensis* (Previously classified as *S. infantarius* subsp. *coli*) in domestic ruminants have been only addressed in caprine and their food products [57,58]. In this study, we also confirmed the presence of *S. lutetiensis* only in the rumen of Korean goats by direct isolation and *sodA* gene-based identification rather that from bovine, thus hypothesizing that the species might be one of the distinguished important commensal SBSEC in caprine. However, due to the limited number of animals and SBSEC isolates in this study, additional studies focusing on the prevalence and characteristics of SBSEC in the rumen of goats by direct bacterial isolation will be necessary in the future.

The 46 *S. equinus* strains were isolated from all three animal species. Although most of the our SBSEC isolates were identified as *S. equinus* based on a *sodA* gene comparison, phylogenetic analyses have indicated that these isolates could be divided as several different types, thus revealing their potential plasticity and diversity and the *S. equinus* isolates were divided as four major types and two individual strains which were not clustered with other isolates, regardless of their source animal species (Figure 1). Among the four major types, *S. equinus* ATCC 9812^T^ and *S. bovis* ATCC 33317^T^ (now synonymized to *S. equinus*) were assigned to type IV and some other bovine-originated SBSEC isolates were clustered together. Therefore, we selected and sequenced the whole genome of the four representative *S. equinus* isolates (CNU_G2, CNU_G3, CNU_G6, and CNU_77-23) to elucidate their plasticity and diversity in the *S. equinus* group. As expected, the genome-based ANI comparison revealed that the genome of *S. equinus* strains CNU_G2 and CNU_G3 showed less than 95.0% ANI values against any of the type strains in the SBSEC including *S. equinus* ATCC 9812^T^. To date, it has been generally accepted that ANI values > 95–96% are the threshold for species delineation [59,60,61] and our results of genome-based ANI comparison also support the potential heterogeneity of *S. equinus*, which has been addressed previously [22]. Similar results were also observed upon the ANI comparison of *S. lutetiensis* CNU_77-61, which showed < 95.0% ANI values against any of the type strains in the SBSEC including *S. lutetiensis* NCTC 13774^T^ even though it was classified as *S. lutetiensis* based on *sodA* gene-based identification. Although only a limited number of domestic ruminants from a single farm and their SBSEC isolates were examined in this study, these results strongly suggest that SBSECs, at least in *S. equinus* and *S. lutetiensis*, are considerably more diverse than previously thought and that the current taxonomic assignments of these two species remains a subject to debate, requiring re-evaluation in the future. Moreover, the three SBSEC strains (CNU_G2, CNU_G3, and CNU_77-61) might be categorized as the potential new species (or subspecies) in the SBSEC and its further studies are currently in progress.

To date, the emergence of AMR bacteria and widespread ARGs has led to serious public health and livestock industry concerns due to potential health risks to humans and animals [23,24]. Furthermore, SBSEC have been reported as one of the most AMR species among streptococci [11]. Although most previous studies evaluating AMR in SBSEC have been focused on isolates of clinical origin, they have clearly indicated high AMR incidence rates against several classes of commercial antibiotics (e.g., clindamycin, erythromycin, and tetracycline) especially in *S. gallolyticus* subsp. *galloyticus*, *S. gallolyticus* subsp. *pasteurianus*, *S. infantarius* subsp. *infantarius*, and *S. lutetiensis* [22,34,35,62,63,64]. These results suggest that SBSEC could be a potential reservoir of various ARGs that can be transmitted to other opportunistic pathogens and other strains in the complex. However, only a few studies are available on AMR SBSEC isolates from animal origins, focusing on *S. gallolyticus*, and these present variable resistance rates [28,29]. The incidence of AMR and ARGs in *S. equinus* and *S. lutetiensis* of an animal origin has not yet been investigated. In this study, the AMR rates of SBSEC isolated from Korean domestic ruminants were relatively lower than those from other SBSEC isolates of clinical origin and the resistance rates on erythromycin, clindamycin, and tetracycline were estimated to be 2.0% (1/51), 0% (0/51), and 15.7% (8/51), respectively. Moreover, the resistance rates of our SBSEC isolates against the three antibiotics were much lower than those reported previously in various streptococcal species including incompletely classified *S. bovis* isolated from milk samples in Korea [27,49]. Along with the phenotypic evaluations of AMRs in SBSEC isolates, we also investigated the presence of ARGs associated with resistance to erythromycin, clindamycin, and tetracycline. Several recent reports have indicated that the dominant erythromycin resistance-associated gene in various streptococcal species isolated from domestic animals was *erm(B)* [65,66] and that resistance in the SBSEC of human clinical origins was mostly associated with *erm(B)* and *mef(A)* [28,29,45,62,67]. However, as expected, *erm(A)*, *erm(B)*, *erm(C)*, and *mef(A)* were not detected in this study, and only the erythromycin-resistant *S. equinus* strain CNU_77-43 might have another unveiled AMR mechanism against the antibiotic. Although the *erm* variants are mostly associated with cross-resistance to macrolides-lincosamides-type B streptogramins (MLS_B_), the bacteria in this genus also contain genes associated with specific resistance to lincosamides, which are commonly used in dairy cattle to treat mastitis [66]. The genes responsible for resistance to lincosamides are *lnu* class genes, which encode nucleotidyltransferases and cause enzymatic inactivation of the antibiotic [68]. Among the nine *lnu* class genes reported to date (http://faculty.washington.edu/marilynr/), the *lnu(C)* is relatively little investigated however, recent reports have demonstrated its presence in the livestock industry [69]. In this study, we confirmed the presence of *lnu(C)* during genome analysis in some SBSEC isolates (strain CNU_77-23 and CNU_G3) even though they were not phenotypically resistant to lincosamides (clindamycin). Therefore, we have screened the presence of *lnu(C)* in all isolates, furthermore, the results also indicated the considerable appearance rate (20.0%) of the genetic determinants although the strains did not show phenotypic resistance to the antibiotics. These phenomena showing a lack of correlation between the presence of ARGs and degree of antimicrobial susceptibility in bacteria have been described previously [66,70,71,72] and thus, further studies on the mechanisms of AMR SBSEC presenting in domestic ruminants are urgently needed.

Different from the results on erythromycin and clindamycin, the phenotypical tetracycline resistance of the *S. equinus* isolates was generally correlated with the presence of *tet(M)* and *Tn916-like transposon* genes except for one strain (CNU_77-29), which showed a low MIC value (0.50 μg/mL) (Table 4). In general, the *tet(M)* gene has been reported to be associated with the conjugative transposons Tn916, Tn1545, and other related transposons [73] and until recently, this gene was mainly considered to have originated from human-originated *S. agalactiae* [74]. However, several recent studies on the surveillance of *tetR* genes in various streptococcal species isolated from domestic animals have reported the frequent presence of *tet(M)* genes in the United States, China, and Europe [66,70,72]. In this study, we only detected the presence of Tn916-like transposon-mediated *tet(M)* genes among the *S. equinus* isolates even though the emergence of several other *tetR* genes (e.g., *tet(L)* and *tet(O)*) has been reported from other SBSEC isolates [28,45,62]. Based on these results, although only a limited number of SBSEC isolates was examined in this study, the protection of ribosomal proteins by *tet(M)* might be one of the main tetracycline-resistance mechanisms in the Korean SBSEC. Additionally, although vancomycin resistance in other SBSEC isolates from animal fecal samples has been rarely reported and its resistance mechanisms were mainly associated with the *vanB* gene [15], a large proportion of SBSEC isolates was categorized as non-susceptible against the antibiotic however, the two groups of *van* resistance operons (D-Ala-D-Lac ligase (*vanA*, *vanB*, *vanD*, and *vanM*) and D-Ala-D-Ser ligase (*vanC*, *vanE*, *vanG*, *vanL*, and *vanN*)) were not detected during our manual search in the five sequenced genomes. Therefore, the phenotypical resistance in our SBSEC isolates might be assumed to be associated with other unknown genetic determinants or point mutations, and further research uncovering its resistance mechanisms will be necessary in the future.

Despite raising concerns regarding their potential pathogenicity and carcinogenic properties, some SBSEC members with a dairy origin have been directly supplied as probiotics to young domestic ruminants to support ruminal development by establishing an anaerobic rumen microbiota for efficient feed digestion [75,76,77,78]. However, the fact that ARGs were found in several SBSEC isolates from different geographical origins is worrying as the bacteria in domestic ruminant could be a potential reservoir of these genes and horizontal gene transfer via transposon within the same genus may lead to serious concerns in the global livestock industry, including Korea. Undeniably, SBSEC are one of the most important commensal rumen bacterial species affecting the health and productivity of domestic animals. Although only a limited number of domestic ruminants from a single farm and their SBSEC isolates were examined, the presence of ARGs, which can potentially be transmitted to other opportunistic pathogens in the animals, was verified in this study. Therefore, nation-wide continuous monitoring of AMR strain emergence and ARG acquisition in SBSEC will be needed in Korean domestic ruminants.

## Figures and Tables

**Figure 1 microorganisms-09-00098-f001:**
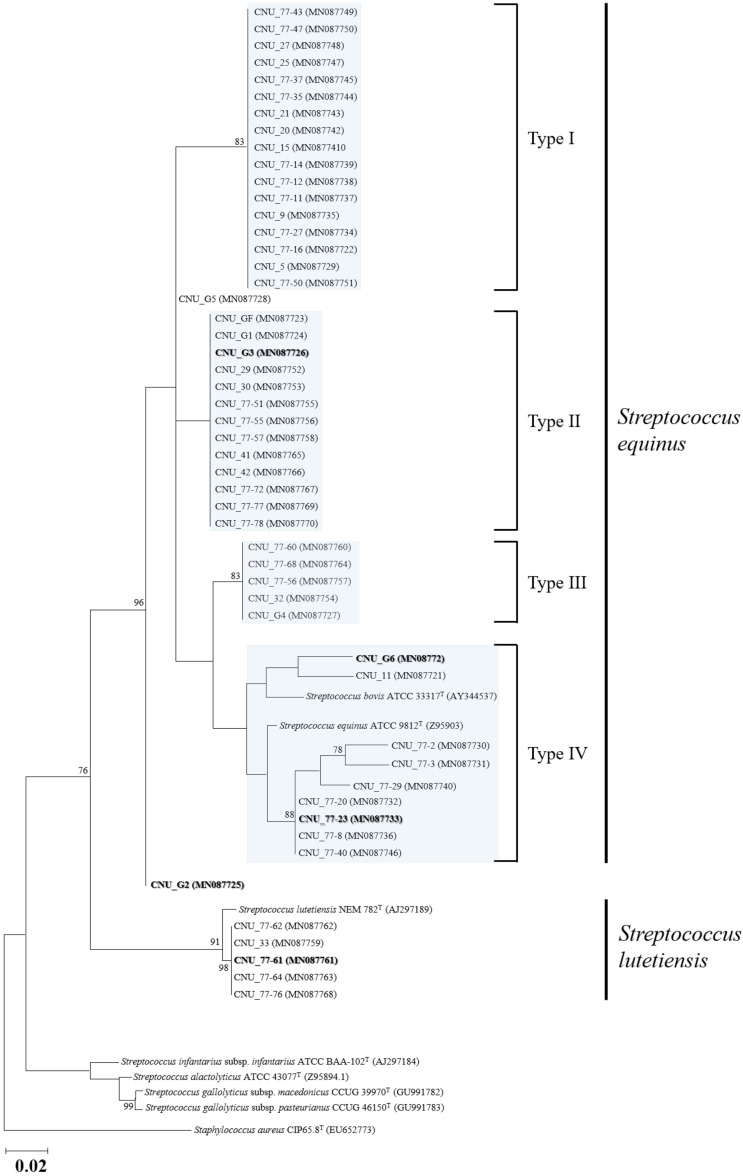
Maximum-likelihood phylogenetic tree based on *sodA* nucleotide sequences showing the relationships of all *Streptococcus* isolates reported in this study with seven representative *sodA* sequences in SBSEC and the outgroup *Staphylococcus aureus* CIP65.8^T^. The blue square boxes denote the presence of four major types (type I, II, III, and IV) of *S. equinus* isolates obtained in this study. The scale bar represents 0.02 nucleotide substitutions per site. The five representative genomes are emphasized in bold.

**Figure 2 microorganisms-09-00098-f002:**
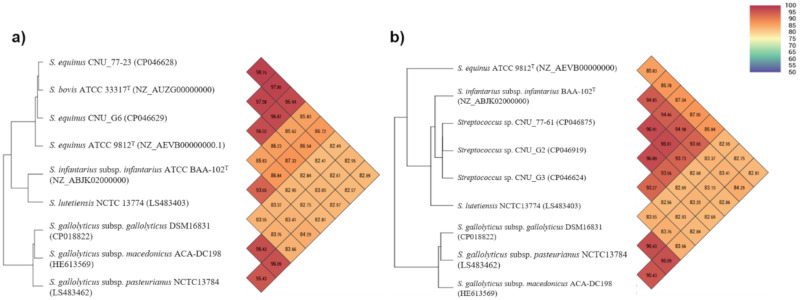
Overall genome relatedness and heatmaps generated with the OrthoANI values calculated using the complete sequenced genomes of *S. equinus* strains CNU_77-23 and CNU_G6 (**a**) and the other three SBSEC strains CNU_77-61, CNU_G2, and CNU_G3 (**b**) compared with other sequenced genomes of SBSEC strains available in the GenBank database. The results of each two-strain comparison are given where the diagonals departing from each strain meet, e.g., the OrthoANI value between *S. equinus* CNU_77-23 and *S. bovis* ATCC 33317^T^ is 97.85%. (2-column fitting image).

**Table 1 microorganisms-09-00098-t001:** *S. bovis*/*S. equinus* complex (SBSEC) isolates obtained in this study.

No.	Bacterial Strains	Host	Isolate Year	16S rRNA Nos. *	*sod*A Gene Nos. *	Deposition No. **
1	*Streptococcus equinus* CNU_5	*Bos taurus*	2014	MN075416	MN087729	KCCM 90360
2	*Streptococcus equinus* CNU_77-2	*Bos taurus*	2014	MN075447	MN087730	KCCM 90361
3	*Streptococcus equinus* CNU_77-3	*Bos taurus*	2014	MN075448	MN087731	KCCM 90362
4	*Streptococcus equinus* CNU_11	*Bos taurus*	2014	MN075419	MN087721	KCCM 90363
5	*Streptococcus equinus* CNU_77-16	*Bos taurus*	2014	MN075450	MN087722	KCCM 90364
6	*Streptococcus equinus* CNU_77-20	*Bos taurus*	2014	MN075451	MN087732	KCCM 90365
7	*Streptococcus equinus* CNU_77-23	*Bos taurus*	2014	MN075454	MN087733	KCCM 90366
8	*Streptococcus equinus* CNU_77-27	*Bos taurus*	2014	MN075455	MN087734	KCCM 90367
9	*Streptococcus equinus* CNU_GF	*Bos taurus*	2018	MN075408	MN087723	KCCM 90354
10	*Streptococcus equinus* CNU_G1	*Bos taurus*	2019	MN075409	MN087724	KCCM 90381
11	*Streptococcus equinus* CNU_G2	*Bos taurus*	2019	MN075410	MN087725	KCCM 90355
12	*Streptococcus equinus* CNU_G3	*Bos taurus*	2019	MN075411	MN087726	KCCM 90356
13	*Streptococcus equinus* CNU_G4	*Bos taurus*	2019	MN075412	MN087727	KCCM 90357
14	*Streptococcus equinus* CNU_G5	*Bos taurus*	2019	MN075413	MN087728	KCCM 90358
15	*Streptococcus equinus* CNU_G6	*Bos taurus*	2019	MN075414	MN087720	KCCM 90359
16	*Streptococcus equinus* CNU_9	*Bos taurus coreanae*	2014	MN075423	MN087735	KCCM 90368
17	*Streptococcus equinus* CNU_77-8	*Bos taurus coreanae*	2014	MN075456	MN087736	KCCM 90369
18	*Streptococcus equinus* CNU_77-11	*Bos taurus coreanae*	2014	MN075459	MN087737	KCCM 90370
19	*Streptococcus equinus* CNU_77-12	*Bos taurus coreanae*	2014	MN075460	MN087738	KCCM 90371
20	*Streptococcus equinus* CNU_77-14	*Bos taurus coreanae*	2014	MN075461	MN087739	KCCM 90372
21	*Streptococcus equinus* CNU_77-29	*Bos taurus coreanae*	2014	MN075462	MN087740	KCCM 90373
22	*Streptococcus equinus* CNU_15	*Bos taurus coreanae*	2014	MN075424	MN087741	KCCM 90374
23	*Streptococcus equinus* CNU_20	*Bos taurus coreanae*	2014	MN075427	MN087742	KCCM 90375
24	*Streptococcus equinus* CNU_21	*Bos taurus coreanae*	2014	MN075428	MN087743	KCCM 90376
25	*Streptococcus equinus* CNU_77-35	*Bos taurus coreanae*	2014	MN075465	MN087744	KCCM 90377
26	*Streptococcus equinus* CNU_77-37	*Bos taurus coreanae*	2014	MN075467	MN087745	KCCM 90378
27	*Streptococcus equinus* CNU_77-40	*Bos taurus coreanae*	2014	MN075468	MN087746	KCCM 90379
28	*Streptococcus equinus* CNU_25	*Bos taurus coreanae*	2014	MN075431	MN087747	KCCM 90380
29	*Streptococcus equinus* CNU_27	*Bos taurus coreanae*	2014	MN075433	MN087748	KCCM 90382
30	*Streptococcus equinus* CNU_77-43	*Bos taurus coreanae*	2014	MN075470	MN087749	KCCM 90383
31	*Streptococcus equinus* CNU_77-47	*Bos taurus coreanae*	2014	MN075473	MN087750	KCCM 90384
32	*Streptococcus equinus* CNU_77-50	*Bos taurus coreanae*	2014	MN075475	MN087751	KCCM 90385
33	*Streptococcus equinus* CNU_29	*Capra aegagrus hircus*	2014	MN075435	MN087752	KCCM 90386
34	*Streptococcus equinus* CNU_30	*Capra aegagrus hircus*	2014	MN075436	MN087753	KCCM 90387
35	*Streptococcus equinus* CNU_32	*Capra aegagrus hircus*	2014	MN075438	MN087754	KCCM 90388
36	*Streptococcus equinus* CNU_77-51	*Capra aegagrus hircus*	2014	MN075476	MN087755	KCCM 90389
37	*Streptococcus equinus* CNU_77-55	*Capra aegagrus hircus*	2014	MN075478	MN087756	KCCM 90390
38	*Streptococcus equinus* CNU_77-56	*Capra aegagrus hircus*	2014	MN075479	MN087757	KCCM 90391
39	*Streptococcus equinus* CNU_77-57	*Capra aegagrus hircus*	2014	MN075480	MN087758	KCCM 90392
40	*Streptococcus equinus* CNU_77-60	*Capra aegagrus hircus*	2014	MN075481	MN087760	KCCM 90394
41	*Streptococcus equinus* CNU_77-68	*Capra aegagrus hircus*	2014	MN075488	MN087764	KCCM 90398
42	*Streptococcus equinus* CNU_41	*Capra aegagrus hircus*	2014	MN075445	MN087765	KCCM 90399
43	*Streptococcus equinus* CNU_42	*Capra aegagrus hircus*	2014	MN075446	MN087766	KCCM 90400
44	*Streptococcus equinus* CNU_77-72	*Capra aegagrus hircus*	2014	MN075490	MN087767	KCCM 90401
45	*Streptococcus equinus* CNU_77-77	*Capra aegagrus hircus*	2014	MN075492	MN087769	KCCM 90403
46	*Streptococcus equinus* CNU_77-78	*Capra aegagrus hircus*	2014	MN075493	MN087770	KCCM 90404
47	*Streptococcus lutetiensis* CNU_33	*Capra aegagrus hircus*	2014	MN075439	MN087759	KCCM 90393
48	*Streptococcus lutetiensis* CNU_77-61	*Capra aegagrus hircus*	2014	MN075482	MN087761	KCCM 90395
49	*Streptococcus lutetiensis* CNU_77-62	*Capra aegagrus hircus*	2014	MN075483	MN087762	KCCM 90396
50	*Streptococcus lutetiensis* CNU_77-64	*Capra aegagrus hircus*	2014	MN075485	MN087763	KCCM 90397
51	*Streptococcus lutetiensis* CNU_77-76	*Capra aegagrus hircus*	2014	MN075491	MN087768	KCCM 90402

* Descriptions of the species of SBSEC isolates in Table 1 were based on the 16S rRNA and *sodA* gene sequence comparisons. ** KCCM, Korean Culture Center of Microorganisms.

**Table 2 microorganisms-09-00098-t002:** List of antimicrobial resistance (AMR)-related genes and *ldh* genes detected in the SBSEC isolates in this study *.

Bacterial Strain	Tetracycline					Lincosamides	MLSb **				*Ldh* ***
	*tet*(M)	*tet(O)*	*tet(Q)*	*tet(S)*	Tn916-like transposase	*lnu(C)*	*erm(A)*	*erm(C)*	*erm(B)*	*mef(A)*	
*Streptococcus equinus*											
CNU_5	-	-	-	-	-	-	-	-	-	-	+
CNU_77-2	+ (MT949160)	-	-	-	+ (MT949172)	-	-	-	-	-	+
CNU_77-3	+ (MT949161)	-	-	-	+ (MT949173)	-	-	-	-	-	+
CNU_11	+ (MT949162)	-	-	-	+ (MT949174)	-	-	-	-	-	+
CNU_77-16	-	-	-	-	-	-	-	-	-	-	+
CNU_77-20	+ (MT949163)	-	-	-	+ (MT949175)	-	-	-	-	-	+
CNU_77-23	+ (MT949164)	-	-	-	+ (MT949176)	+ (MT949150)	-	-	-	-	+
CNU_77-27	-	-	-	-	-	-	-	-	-	-	+
CNU_GF	+ (MT949165)	-	-	-	+ (MT949177)	+ (MT949151)	-	-	-	-	+
CNU_G1	+ (MT949166)	-	-	-	+ (MT949178)	+ (MT949152)	-	-	-	-	+
CNU_G2	+ (MT949167)	-	-	-	+ (MT949179)	-	-	-	-	-	+
CNU_G3	-	-	-	-	-	+ (MT949153)	-	-	-	-	+
CNU_G4	-	-	-	-	-	+ (MT949154)	-	-	-	-	+
CNU_G5	+ (MT949168)	-	-	-	+ (MT949180)	-	-	-	-	-	+
CNU_G6	+ (MT949169)	-	-	-	+ (MT949181)	-	-	-	-	-	+
CNU_9	-	-	-	-	-	-	-	-	-	-	+
CNU_77-8	-	-	-	-	-	+ (MT949155)	-	-	-	-	+
CNU_77-11	-	-	-	-	-	-	-	-	-	-	+
CNU_77-12	-	-	-	-	-	-	-	-	-	-	+
CNU_77-14	-	-	-	-	-	-	-	-	-	-	+
CNU_77-29	+ (MT949170)	-	-	-	+ (MT949182)	-	-	-	-	-	+
CNU_15	-	-	-	-	-	-	-	-	-	-	+
CNU_20	-	-	-	-	-	-	-	-	-	-	+
CNU_21	-	-	-	-	-	-	-	-	-	-	+
CNU_77-35	-	-	-	-	-	-	-	-	-	-	+
CNU_77-37	-	-	-	-	-	-	-	-	-	-	+
CNU_77-40	+ (MT949171)	-	-	-	+ (MT949183)	-	-	-	-	-	+
CNU_25	-	-	-	-	-	-	-	-	-	-	+
CNU_27	-	-	-	-	-	-	-	-	-	-	+
CNU_77-43	-	-	-	-	-	-	-	-	-	-	+
CNU_77-47	-	-	-	-	-	-	-	-	-	-	+
CNU_77-50	-	-	-	-	-	-	-	-	-	-	+
CNU_29	-	-	-	-	-	-	-	-	-	-	+
CNU_30	-	-	-	-	-	-	-	-	-	-	+
CNU_32	-	-	-	-	-	+ (MT949156)	-	-	-	-	+
CNU_77-51	-	-	-	-	-	-	-	-	-	-	+
CNU_77-55	-	-	-	-	-	-	-	-	-	-	+
CNU_77-56	-	-	-	-	-	+ (MT949157)	-	-	-	-	+
CNU_77-57	-	-	-	-	-	-	-	-	-	-	+
CNU_77-60	-	-	-	-	-	+ (MT949158)	-	-	-	-	+
CNU_77-68	-	-	-	-	-	+ (MT949159)	-	-	-	-	+
CNU_41	-	-	-	-	-	-	-	-	-	-	+
CNU_42	-	-	-	-	-	-	-	-	-	-	+
CNU_77-72	-	-	-	-	-	-	-	-	-	-	+
CNU_77-77	-	-	-	-	-	-	-	-	-	-	+
CNU_77-78	-	-	-	-	-	-	-	-	-	-	+
*Streptococcus lutetiensis*											
CNU_33	-	-	-	-	+ (MT949184)	-	-	-	-	-	+
CNU_77-61	-	-	-	-	+ (MT949185)	-	-	-	-	-	+
CNU_77-62	-	-	-	-	+ (MT949186)	-	-	-	-	-	+
CNU_77-64	-	-	-	-	+ (MT949187)	-	-	-	-	-	+
CNU_77-76	-	-	-	-	+ (MT949188)	-	-	-	-	-	+

* All sequenced and confirmed AMR-related genes and *ldh* genes were deposited in the GenBank database. ** MLS_b_: macrolide-lincosamide-streptogramin group. *** ldh: L(+)-lactate dehydrogenase gene.

**Table 3 microorganisms-09-00098-t003:** Antimicrobial-resistance profile of 51 SBSEC isolates in this study.

Strains		Antimicrobial Agents [μg (Disks) or mg/L (MIC)]										
	Am *	Carb	Cep	Fq	Lin *	Ma *	Penicillins		Phe	Tet *	Gly	Oxa
	CN (0.016–256)	IPM (10)	KF (30)	LEV (5)	CD (0.016–256)	E (0.016–256)	OX (1)	P (10)	C (30)	TE (0.016–256)	VA (30)	LZD (30)
*Streptococcus equinus*												
CNU_5	3				0.094	0.19				3		
CNU_77-2	2				0.064	0.047				12		
CNU_77-3	1.5				0.047	0.50				16		
CNU_11	5				0.064	0.19				4		
CNU_77-16	12				0.064	0.19				4		
CNU_77-20	4				0.064	0.50				32		
CNU_77-23	16				1	0.25				32		
CNU_77-27	12				0.094	0.25				3		
CNU_GF	12				0.125	0.19				6		
CNU_G1	12				0.125	0.094				32		
CNU_G2	16				0.125	0.19				12		
CNU_G3	6				0.5	0.19				6		
CNU_G4	16				0.047	0.19				0.75		
CNU_G5	8				0.064	0.064				3		
CNU_G6	4				0.094	0.094				32		
CNU_9	24				2.5	0.50				3		
CNU_77-8	4				0.25	0.125				4		
CNU_77-11	24				0.094	0.38				3		
CNU_77-12	8				0.094	0.38				4		
CNU_77-14	8				0.19	0.38				4		
CNU_77-29	3				0.064	0.094				0.50		
CNU_15	12				0.094	0.50				3		
CNU_20	8				0.125	0.19				2		
CNU_21	8				0.19	0.50				3		
CNU_77-35	32				0.19	0.50				3		
CNU_77-37	12				0.25	0.19				4		
CNU_77-40	4				0.094	0.19				48		
CNU_25	8				0.125	0.75				2		
CNU_27	12				0.125	0.50				3		
CNU_77-43	24				0.38	6				6		
CNU_77-47	24				0.094	0.50				3		
CNU_77-50	12				0.125	0.50				3		
CNU_29	6				0.094	0.19				2		
CNU_30	6				0.125	0.25				0.5		
CNU_32	6				0.125	0.19				0.75		
CNU_77-51	6				0.094	0.19				0.5		
CNU_77-55	8				0.094	0.25				0.5		
CNU_77-56	12				0.19	0.19				0.5		
CNU_77-57	16				0.125	0.0125				3		
CNU_77-60	24				0.125	0.25				3		
CNU_77-68	48				0.125	0.19				1		
CNU_41	6				0.047	0.25				3		
CNU_42	8				0.125	0.125				1		
CNU_77-72	8				0.094	0.125				3		
CNU_77-77	6				0.094	0.032				0.50		
CNU_77-78	8				0.064	0.125				0.38		
*Streptococcus lutetiensis*												
CNU_33	12				0.125	0.032				1		
CNU_77-61	8				0.19	0.25				3		
CNU_77-62	12				0.064	0.19				3		
CNU_77-64	8				0.094	0.25				2		
CNU_77-76	5				0.19	0.25				3		
Susceptible	15.7% (8/51)	100% (51/51)	90.2% (46/51)	0% (0/51)	94.1% (48/51)	72.5% (37/51)	98.0% (50/51)	9.8% (5/51)	3.9% (2/51)	29.4% (15/51)	15.7% (8/51)	5.9% (3/51)
Intermediate	62.7% (32/51)	0% (0/51)	9.8% (5/51)	76.5% (39/51)	5.9% (3/51)	25.5% (13/51)	2.0% (1/51)	90.2% (46/51)	47.1% (24/51)	54.9% (28/51)	84.3% ** (43/51)	94.1% ** (48/51)
Resistant	21.6% (11/51)	0% (0/51)	0% (0/51)	23.5% (12/51)	0% (0/51)	2.0% (1/51)	0% (0/51)	0% (0/51)	49.0% (25/51)	15.7% (8/51)		

The category of antibiotic susceptibility is indicated as follows: Dark gray, resistant; light gray, intermediate; white, susceptible. Am, Aminoglycosides; CN, Gentamicin; Carb, Carbapenems; IPM, Imipenem; Cep, Cephalosporins; KF, Cephalothin; Fq, Fluoroquinolones; LEV, Levofloxacin; Lin, Lincosamides; CD, Clindamycin; Ma, Macrolides; E, Erythromycin; OX, Oxacillin; P, Penicillin; Phe, Phenicols; C, Chloramphenicol; Tet, Tetracyclines; TE, Tetracycline; Gly, Glycopeptides; VA, Vancomycin; Oxa, Oxazolidinones; LZD, Linezolid. * The results of MIC evaluations are shown in numbers. ** According to the CLSI guidelines, the results of susceptibilities against vancomycin and linezolid in Viridans group Streptococci were referred as susceptible and non-susceptible (replacing intermediate and resistant).

**Table 4 microorganisms-09-00098-t004:** The genomic features of five selected SBSEC isolates.

Feature	Strains				
	CNU_77-23 (CP046628)	CNU_77-61 (CP046875)	CNU_G2 (CP046919)	CNU_G3 (CP046624)	CNU_G6 (CP046629)
Size (bp)	1,911,874	1,917,833	1,960,491	1,864,202	1,910,720
G + C content (%)	37.4	37.8	37.9	37.8	37.4
Conitgs	1	1	2	2	2
plasmids	-	-	1	1	1
Total genes	1883	1910	2056	1885	1948
tRNAs	70	70	70	69	70
rRNAs	21	21	21	21	21
ncRNAs	4	4	4	4	4
Protein-coding genes	1771	1778	1926	1759	1834
Pseudogenes	17	37	35	32	19

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
