# Peer review of "Diversity and Antimicrobial Resistance in the Streptococcus bovis/Streptococcus equinus Complex (SBSEC) Isolated from Korean Domestic Ruminants"

_microorganisms, 2021, doi:10.3390/microorganisms9010098_

Round 1

Reviewer 1 Report

Review of Manuscript Microorganisms-1032769

The paper studied the diversity and AMR of the SBSE complex from Korean domestic ruminants. In principle the paper is innovative and interesting, used novel and sophisticated methods, methods and materials were very well described and I am confident that authors performed well the experiment. The writing style is also very good easy to read and follow. Authors could scientifically discuss their findings or provide readers with interesting interpretations of results. However, I think the paper have few little weaknesses and I recommend authors to consider the following remarks to improve the quality of the manuscript:

In general, the introduction brings the problem in a general context but I am still missing a good justification of this study. What is the importance of characterizing the rumen bacteria diversity of ruminants? What is the importance of studying the AMR or the antimicrobial susceptibility of the SBSE complex? What would be interesting for ruminant scientists and nutritionists? Authors must bring into discussion also the practical use of these questions and of their possible findings. Otherwise, the topic is not worth for study. I think these questions were not clearly addressed in the justification and must be considered. Or optionally, can be considered in the discussion chapter

Material and Methods: How many animals per species were sampled? How many per year? Were the same animals each year? Were samples somehow pooled? How can you assure that number of animals samples can be enough representative? What kind of diets received the animals? Detailed information is here required for interpretation of results. Moreover, animals sampled only belonged to one farm. How can we assure that the diversity and the AMR here found can be extrapolated to whole Korea? I think this is one of the biggest gaps of the present study. This limitation must be mentioned and possible consequences for interpretations must be addressed.

Specific comments

L312: I am not agree that they cause “directly” rumen acidosis. Rumen acidosis can be only caused if starch is fed in high quantities. Rewrite the sentence

L316: Metabolic disorders is too general. Be specific

L321: Are these results limited only to animals from Korea or can be generalized? Why not?

L347: Was the S lutetiensis reported in goats in other regions of the world?

Author Response

Please see the attached file. Thank you for advice and comments.

Happy new year.

Reviewer 2 Report

As one of the most important metabolic disorders in intensive ruminants, acute rumen lactic acidosis could be caused by the lactic acid-producing bacteria in the Streptococcus bovis/equinus complex (SBSEC). Park et al. here investigated the diversity of SBSEC in Korean domestic ruminants and studied the antimicrobial resistance genes in the isolated identified in this study by bacterial isolation, species phylogenetic analysis, antimicrobial susceptibility test, whole genomic sequencing and PCRs determination of antimicrobial-resistant genes. Here are some concerns should be addressed:

  1. Lines 20 and 77: Please remove the description of “mechanisms”. There was no mechanism study and data related in the text, the detection of AMR associated genes was not enough for the mechanism.
  2. Lines 24-27: Please re-write this sentence.
  3. Lines 63-72: Some background of antimicrobial resistance associated genes may be included here.
  4. Line 96: How the 16S rRNA sequencing analysis was performed? PCR amplification and Sequencing? More details should be clarified here.
  5. Line 105: Was the sequencing performed with PCR products, or with the gel purification after running the gel electrophoresis with the PCR products? Also, the information of the PCR running kit and sequencing company should be included in the text.
  6. Lines 114-129: The negative control was also included in the study, right? How many repeats were performed in the study?
  7. Lines 178-180: Were there some isolates from the same animal? No single animal with more than one bacterial isolate?
  8. Lines 186-204: Why only one reference sequence was included in the phylogenetic tree analysis? More reference strains should be included, especially strains from Korea (if possible), or source counties close Korea.

      Why not include the reference sequence of S. bovis here? Since the stains isolated in this study were quite different with the S. equinus references.

      How about the polygenetic analysis with the 16S rRNA gene? Was the similar result obtained with sodA gene?

  1. Figure 2: Please include the scale bar inside the figure and figure legend.

Author Response

(The authors gave the same response as above.)

Round 2

Reviewer 2 Report

The authors addressed all the concerns. I agree to published in current version.